# Influence of Single/Collective Use of Curing Agents on the Curing Behavior and Bond Strength of Soy Protein-Melamine-Urea-Formaldehyde (SMUF) Resin for Plywood Assembly

**DOI:** 10.3390/polym11121995

**Published:** 2019-12-02

**Authors:** Zhigang Wu, Bengang Zhang, Xiaojian Zhou, Lifen Li, Liping Yu, Jingjing Liao, Guanben Du

**Affiliations:** 1College of Forestry, Guizhou University, Guiyang 550025, China; wzhigang9@163.com (Z.W.); lifenli2011@163.com (L.L.); ylpgzu@163.com (L.Y.); 2Yunnan Provincial Key Laboratory of Wood Adhesives and Glued Products, Southwest Forestry University, Kunming 650224, China; zbg18082968142@163.com (B.Z.); jingjingliao0915@gmail.com (J.L.)

**Keywords:** soy protein-melamine-urea-formaldehyde (SMUF) resin, curing agent, curing and thermal behavior, shear strength

## Abstract

Soybean protein hydrolysate, melamine, urea, and concentrated formaldehyde were used to synthesize an environmentally friendly soybean protein-melamine-urea-formaldehyde (SMUF) co-condensation resin. (NH_4_)_2_SO_4_, (NH_4_)_2_HPO_4_, (NH_4_)_2_HPO_4_ + (NH_4_)_2_SO_4_, (NH_4_)_2_HPO_4_ + (NH_4_)_2_S_2_O_8_, and (NH_4_)_2_HPO_4_ + (NH_4_)_2_SO_4_ + (NH_4_)_2_S_2_O_8_ were employed as curing agents for SMUF resin. The curing and thermal behaviors of the SMUF resin were investigated using differential scanning calorimetry (DSC), thermogravimetric analysis (TGA), and scanning electron microscopy (SEM). The results revealed the following: (1) (NH_4_)_2_SO_4_ alone could not cure the SMUF resin completely; thus, the final shear strength accomplished plywood with the resin was low, and its water resistance was poor, while the adhesive section was loose and porous/brittle after curing. (2) (NH4)2HPO4 could be hydrolyzed to generate H^+^ and promote SMUF curing, but it could also form polyphosphoric acids, resulting in crosslinking reactions with SMUF in parallel; thereby, the curing properties were improved. (3) When (NH_4_)_2_HPO_4_ + (NH_4_)_2_SO_4_ + (NH_4_)_2_S_2_O_8_ were engaged collectively as curing agent, the shear strength, water resistance, and heat resistance of SMUF attained were the best possible whereas the curing temperature was decreased and the heat released by curing was elevated substantially, which signifies maximized extent of crosslinking was achieved. Further, the adhesive section exhibited mostly a crosslinking intertexture as demonstrated by means of SEM. Accordingly, this study may serve as a guide for the curing of amino resins, with low-molar ratio of formaldehyde to amine in adhesives, which are applied to plywood production.

## 1. Introduction

Great interest is being dedicated to the modification and applications of melamine-urea-formaldehyde (MUF) resin adhesive in wood panel industry, such as particleboard (PB), plywood and medium density fiberboards (MDF), due to the excellent application prospects of this co-condensation resin [1,2,3]. However, the high brittleness after curing, short pot life, free formaldehyde emission, and high melamine cost are major drawbacks that strongly restrict the application and use of this adhesive system. As such, researchers and production enterprises continued to seek ways to modify the adhesive [4,5,6,7,8]. Various modification methods exist currently, among which biomass-based co-condensation technology has drawn much attention. This technology made considerable use of renewable raw materials bearing in mind the resin properties are guaranteed. 

Biomass raw materials, such as soybean protein, tannin and starch, are either blended with MUF resin directly before the preparation of plywood [9,10,11] or used to form biomass-based polymerization resin systems together with MUF [12,13,14,15,16]. These two methods brought about ideal modification effects as they not only control the cost of resin synthesis and reduce free formaldehyde emissions but also improve the properties of the resulting MUF resin adhesives in such a way that the derived wood panels meet the standard requirements. 

Managing a balance between formaldehyde emission and maximum best physicochemical properties of amino resins as wood adhesives is a crucial matter and consequently became a foremost task for researchers [17,18,19]. Decreasing the formaldehyde molar ratio in these resins is among the approaches dedicated for controlling the formaldehyde emission from wood panels. Regrettably, there was a negative impact on the physical and mechanical properties, but even more, the curing rate was slowed down significantly leading to low production line efficiency [20]. Meanwhile, traditional and single curing agents cannot easily solve all the drawbacks of co-condensation resin systems. 

The current study investigates a prospect for managing the required balance with the use of collective curing systems for soybean protein-melamine-urea-formaldehyde (SMUF) resin and the impact of this on the curing behavior and end-product properties. 

## 2. Materials and Methods

### 2.1. Materials

Soybean protein hydrolysate with a viscosity of 32 mPa·s and solid content of 36% was prepared in a laboratory scale [16]. Formaldehyde solution with 50 wt % concentration was provided by Kunming Xinfeilin Panel Board Co., Ltd. (Kunming, China). Poplar (Populus euramevicana cv.”I-214”) veneer with the dimensions 400 mm × 400 mm × 2 mm and 8% to 10% moisture content was purchased from Qunyou Wood Co., Ltd. (Linyi, China) for plywood preparation. Other chemicals were purchased from Sinopharm Chemical Reagent Co., Ltd. (Shanghai, China).

### 2.2. Preparation of Soybean Protein-Melamine-Urea-Formaldehyde Resin Adhesive

Soy protein-melamine-urea-formaldehyde (SMUF) resin adhesive was prepared as follows: Briefly, 97.8 parts of 50 wt % formaldehyde solution was charged to a three-necked 500 mL flask equipped with a mechanical stirrer, thermometer, and condenser. The pH was adjusted to 9.0–9.2 with 30% NaOH solution, and the temperature was raised gradually. After reaching 50 °C, a first portion of urea (37.6 parts) and soybean protein hydrolysate (4.6 parts) were charged to the formaldehyde solution under vigorous stirring. Once the temperature reached 90–95 °C, a first portion of melamine (5.95 parts) was added to the mixture, and the solution pH was adjusted to 5.2–5.3 with 10% formic acid. When the viscosity of the mixture attained 90 mPa·s, the solution pH was adjusted to 8.7–8.9, and the temperature was decreased to 86–88 °C. Afterwards, a second portion of melamine (34.725 parts) was added to the mixture. When the water tolerance reached 100%, the solution pH was immediately adjusted to 9.0, and the temperature was decreased to 45 °C. A second portion of urea (8.55 parts) was charged to the system, and the pH was adjusted to 8.0–8.5; then, the resin was cooled to room temperature. Neat MUF resin adhesive with F/(M + U) molar ratio equal to 1.5 was prepared in a similar way to above procedure, with no soybean protein hydrolysate addition, for comparison purposes.

### 2.3. Use of the Curing Agents for SMUF Resin Adhesive

A total of 1.6 wt % curing agent (based on the solid content of the resin) was added and mixed thoroughly to SMUF resin prior to the preparation of the plywood. The composition of the curing agents was formulated in Table 1. 

The pH of the SMUF resin was tested after curing to predict the influence of residual acids on the properties of plywood formulated with SMUF. Typically, cured SMUF (2 g) was mixed with 20 mL of distilled water for 3 min and allowed to settle for 6 h. The mixture was then homogenized once more, and its pH was measured. The reported results are mean of measurements on five samples.

### 2.4. Preparation of Plywood with SMUF and Testing of the Shear Strength

The SMUF resin adhesives were used to prepare three-layer plywood with dimensions of 400 mm × 400 mm × 6 mm. Veneers with a double-sided adhesive loading of 220 g/m^2^ were rested at room temperature for 15–20 min. The assembled veneers were then exposed to single-layer hot press unit (XLB type) at Shanghai Rubber Machinery Plant and pressed with a pressure of 1.0 MPa at 130 °C for 5 min to obtain a plywood panel. Triplicates of three-layer plywood panel were prepared and conditioned in the laboratory at 20 ± 2 °C and relative humidity of 65 ± 5% for 1 day prior to cutting into shear specimens with dimensions of 100 mm × 25 mm. The wet shear strength of the plywood specimens was tested according to Chinese National Standard (GB/T 17657-2013) after subjecting to boiling water. A mechanical testing machine (model WDS-50KN) was used to determine the shear strength of the plywood specimens. The reported wet shear strength is the mean of 8–10 specimens.

### 2.5. Fourier Transform Infrared Spectroscopy

About 0.001 g of sample powder was mixed well with 1 g of KBr to prepare a pellet. The pellet was then subjected to fourier transform infrared spectroscopy (FT-IR; Scimitar 1000, Varian, PALO ALTOUSA, USA), and the data was acquired within the range of 400–4000 cm^−1^ at 4 cm^−1^ resolution using 32 scans.

### 2.6. ^13^C Nuclear Magnetic Resonance Spectroscopy

A liquid sample (400 μL) was dissolved in 200 μL of dimethyl sulfoxide-d6 (DMSO-d6), and the zgig spectra were collected using a Bruker AVANCE spectrometer operated at a frequency of 600 MHz. The chemical shifts were calculated relative to (CH_3_)_3_Si(CH_2_)_3_SO_3_Na, dissolved in DMSO-d6 as a control. The spectra were measured at 39062.5 Hz over 500–800 nt. All tests were run with a relaxation delay of 6 s and the chemical shifts were accurate to 1 ppm. 

### 2.7. Differential Scanning Calorimetry

A Perkin–Elmer differential scanning calorimeter (*DSC* 204F1, Rodgau, Germany) was used for investigation of thermal analysis. A total of 1.6 wt % curing agent was mixed with the resin prior to the DSC analysis. The thermal run for each sample was performed at a heating rate of 10 °C/min. over a temperature range of 30–180 °C. PYRISTM Version 4.0 software (Rodgau, Germany) was used for data treatment. 

### 2.8. Thermogravimetric Analysis

A thermogravimetric (TG) analyzer (NETZSCH; Bavaria, Germany) was used for evaluating the thermal resistance of the samples under nitrogen atmosphere at a heating rate 10 °C/min from room temperature up to 450 °C. 

### 2.9. Scanning Electron Microscopy Imaging

Fractured cross-sections of the cured SMUF were examined using a Hitachi S-3400N emission scanning electron microscope (SEM, Tokyo, Japan) operated at 12.5 kV for investigating the morphological features at the interface.

## 3. Results and Discussion

### 3.1. Structural Analysis of SMUF Resin

Figure 1 shows the FT-IR curves of MUF and SMUF. The peak at 1652.5 cm^−1^ could be attributed to C=O vibrations while that at 1555.9 cm^−1^ indicated bending and ring deformation vibrations of N–C=N in the triazine ring. Further, the band at 1371.5 cm^−1^ belongs to a bending vibration of –CH_2_ [21]. Compared with that of MUF, the collapse of the characteristic peak at 1555.9 cm^−1^ may indicate that soy protein started to partially participate in the network structure at the expense of the melamine triazine ring. This might be further corroborated by the reshaping and broadening of the peak centered at 3400 cm^−1^, matching with complicated overlapping of different N–H and OH environments. 

The peak at 1289.4 cm^−1^ could be attributed to the antisymmetric stretching vibrations of CH_3_O–. Moreover, a peak at 1134.9 cm^−1^ reflects the symmetric stretching vibrations of C–O–C whereas the peak at 1026.7 cm^−1^ belongs to –CH_2_OH. The peak at 898.2 cm^−1^ reflects C–H deformation vibrations between the triazine ring and H outside of the adjacent ring. Further, the peak at 816.6 cm^−1^ indicates the triazine ring out-of-plane vibrations. Thus, the peaks at 1555.9 and 816.6 cm^−1^ are important characteristics of the melamine triazine ring. However, these FT-IR spectra for MUF and SMUF are mostly identical and matching with the expected structure but could not be relied on neither to differentiate both structures nor to confirm active insertion of soy protein to be a permanent part of MUF. 

Figure 2 shows the ^13^C NMR spectra of MUF and SMUF. The peak at 83 ppm belongs to methylene glycol (developed after reaction with formaldehyde). Peaks corresponding to the hydroxymethylation products (C and D) appeared at 63–65 ppm. Peaks at 46–48 ppm (A) and 54–56 ppm (B) mainly indicate methylene bridge bonds, while the peaks at 67–70 ppm are assigned to methylene ether bonds (E); also, the peaks at 74–75 ppm (F) and 77–78 ppm (G) refer to methylene ether bonds.

Hydroxymethylation is the basis and monomeric precursor for obtaining resin with increased molecular chains and liability for crosslinking reactions. Indeed, the hydroxymethyl content of a resin can reflect its polycondensation degree. The higher the hydroxymethyl content, a more thorough is the addition reaction. Methylene bridge and methylene ether bonds are obtained from hydroxymethyl consumption. Therefore, most of the structural skeleton in the resin is made up of these bonds, and the higher the polycondensation degree, the higher the skeleton strength of the resin. Compared with the spectra of MUF in Figure 2, the relevant one for SMUF has a higher methylene bridge bonds content and thereby polycondensation degree [22,23]. In addition, a weak peak at 83.33 ppm could be observed for MUF, but a corresponding peak at this position was not obvious for SMUF; this finding indicates indirectly that the free formaldehyde content of SMUF is lower as compared to MUF. 

Given its potentially high molecular weight, soy protein could abruptly increase the viscosity of an adhesive and generate a gel when directly used as modifying agent for MUF. Soy protein molecules are usually spherical in shape and contain several active groups, such as amino and carboxyl groups. These function groups are mostly hidden within these molecules; therefore, soy protein must be degraded before chemical crosslinking [24,25,26]. The active sites will be only accessible when spherical proteins are degraded into shorter segments, which render them more exposed [21]. These active groups may undergo the following chemical reactions with melamine, urea, and formaldehyde (Figure 3): (1) a reaction between formaldehyde and soy protein, (2) a reaction between hydroxymethyl melamine and soy protein, and (3) a reaction between hydroxymethyl melamine and hydroxymethyl protein. A reaction in which methylene ether links are transformed into methylene bridged bonds is also known to occur in soybean protein and MUF resin system. This reaction can considerably reduce the free formaldehyde content of the resin adhesive. 

### 3.2. Basic Physicochemical Properties of SMUF Resin Adhesive

Table 2 shows the properties of MUF and SMUF resins. The solid contents of SMUF and MUF were sufficiently high; thus, dehydration was not required when these adhesives were used to prepare particleboard and plywood; thus, the viscosity levels were reasonable for application as wood adhesives in both cases as well. Moreover, the SMUF resin adhesive is pale yellow, similar to the color of wood, which is considered to be an advantage as it will not affect wood color when used as an adhesive.

Solution pH is a decisive factor influencing the curing rate of amino resin adhesives. NH_4_Cl is the most commonly used curing agent [1]. This salt catalyzes the curing reaction of MUF resin via reactions (1) and (2), while reaction (1) plays a special dominant role. Therefore, free formaldehyde exerts a critical effect in NH_4_Cl-catalyzed melamine resin curing. Formaldehyde can react with NH_4_Cl to generate HCl, thereby promoting the curing of amino resin adhesives. However, when NH_4_Cl is added as a curing agent, the wood panel generates extremely toxic dioxin during combustion while low-molar ratio urea formaldehyde (UF) and MUF resins cannot be cured completely [27,28]. To overcome this issue, (NH_4_)_2_SO_4_ was used as a curing agent for amino resin adhesives such as UF and MUF.

NH_4_Cl + HCHO → (CH_2_)_6_N_4_ + HCl + H_2_O(1)

NH_4_Cl + H_2_O → NH_4_Cl + NH_4_OH(2)

The properties of SMUF treated with different curing agents are shown in Table 3. (NH_4_)_2_SO_4_ can react with formaldehyde to generate H_2_SO_4_ and promote SMUF curing (CA-1). The shear strength of plywood after exposure to boiling water is only 1.71 MPa. Traditionally prepared MUF resin has free formaldehyde content of 0.2% to 0.5% or a little higher. High free formaldehyde content can ensure a facile curing of MUF. However, the SMUF resin prepared in this study has low free formaldehyde content (0.065%) and sufficient curing to be achieved by (NH_4_)_2_SO_4_ is rather difficult, which accounts for the low shear strength. 

When (NH_4_)_2_HPO_4_ is used for curing (CA-2), the shear strength of plywood after exposure to boiling water increased to 2.03 MPa. The mechanisms for curing SMUF resin using (NH_4_)_2_HPO_4_ and (NH_4_)_2_SO_4_ are quite similar. The generated acids are H_3_PO_4_ (moderate-strength acid) and H_2_SO_4_ (strong acid), respectively. It is therefore expected that the catalytic reaction rate in case of H_3_PO_4_ is slower than that of H_2_SO_4_. However, since (NH_4_)_2_HPO_4_ contains more H^+^ than (NH_4_)_2_SO_4_, the curing process of SMUF proceeded more efficiently, and shear strength was enhanced. Similarly, the hydrolysis of (NH_4_)_2_HPO_4_ at room temperature occurs slowly; the pH change in the resin is minor when (NH_4_)_2_HPO_4_ is directly blended with SMUF resin, and the residual acid in the resin is weak after curing. Therefore, (NH_4_)_2_HPO_4_ is an ideal curing agent for SMUF resin when only one curing agent is of interest. 

When the binary curing agents (NH_4_)_2_HPO_4_ + (NH_4_)_2_SO_4_ (CA-3) and (NH_4_)_2_HPO_4_ + (NH_4_)_2_S_2_O_8_ (CA-4) were used for the curing of SMUF resin, the shear strengths of relevant plywood after immersion in boiling water increased to 2.16 and 2.31 MPa, respectively. This is compiled to an increase in the extent of curing by 26.3% and 35.1%, respectively. Hence, the catalytic achievement of CA-4 on SMUF is superior with respect to that of CA-3. This is attributed to the slow hydrolysis of (NH_4_)_2_S_2_O_8_ in solution at low temperatures, while its hydrolysis rate increases considerably with an increase in the temperature. This is made clearer in Equations (3)–(6) which show the sequenced hydrolytic reactions involving (NH_4_)_2_S_2_O_8_. 

It is worthy to mention that the pH values of the adhesive systems upon using CA-3 and CA-4 as curing agents were 7.01 and 6.32, respectively. The latter system is slightly acidic, which most likely results in partial curing of the resin, poor curing uniformity, and brittleness. In case of SMUF, the curing reaction catalyzed by CA-4 acquired more acidity (pH ~ 5.61). Most chemical reactions are reversible, therefore acidic systems can accelerate the curing of SMUF resin. Residual acids can also accelerate the degradation of the curing products, thereby affecting the usability of the SMUF resin.
(NH_4_)_2_S_2_O_8_ + H_2_O → (NH_4_)_2_SO_4_ + H_2_SO_4_(3)
(NH_4_)_2_S_2_O_8_ + HCHO → (CH_2_)_6_N_4_ + H_2_S_2_O_8_(4)
H_2_S_2_O_8_ + HCHO → H_2_SO_4_ + HCOOH(5)
H_2_S_2_O_8_ + H_2_O → (NH_4_)_2_SO_4_ + H_2_SO_4_(6)

When (NH_4_)_2_HPO_4_ + (NH_4_)_2_SO_4_ + (NH_4_)_2_S_2_O_8_ (CA-5) are used as a collective curing agent for SMUF resin, the shear strength of plywood after soaking in boiling water for 2 h is elevated to 2.58 MPa with an increase extent of 50.9%. This kind of ternary curing agent can realize full curing of the SMUF resin and the influence of any residual acidic species after curing is limited, which benefits the stability. 

Compared to blank SMUF resin (without curing agent addition) in Table 3, single/collective curing agent addition can significantly affect the properties of SMUF resin in viscosity, pH, and mechanical shear strength in boiling water. The most obvious is that the mechanical shear strength increased more than 10 times compared with blank SMUF resin. This fully proved the importance of adding curing agent.

### 3.3. Analysis of the Curing Behavior of SMUF Resin Adhesive

The curing thermograms obtained from DSC runs for SMUF resin adhesives, treated with different curing agents, are shown in Figure 4 along with other corresponding parameters. The curing temperature and associated heat released by the SMUF resins are 111.6 °C and 334.3 mW/mg, respectively, when (NH_4_)_2_SO_4_ was used as the curing agent. When (NH_4_)_2_HPO_4_ was used instead, the SMUF resin curing temperature decreased to 109.9 °C, and the liberated heat as a result of curing increased to 425.7 mW/mg. Again, this is accounted for by (NH_4_)_2_HPO_4_ providing relatively more H^+^ compared with (NH_4_)_2_SO_4_ to promote the curing reaction. This decreased in turn the curing temperature, elevated the degree of curing, and so the amount of heat released increased. 

When CA-3 and CA-4 were used as curing agent, the curing temperatures of SMUF resin reached at 109.1 and 107.9 °C, respectively. The latter temperature is lower because the hydrolysis products of (NH_4_)_2_S_2_O_8_ are acidic and can further promote the SMUF curing. Although CA-4 showed accelerated curing, it has no obvious effect on the heat released during curing, and the heat released was 463.3 mW/mg, which is only slightly higher than that of CA-3 curing system (459.9 mW/mg). This finding is accounted for by the heat needed in parallel for the decomposition of the components of this curing system (Equations (3)–(6)). 

When the triple-component system (CA-5) was used as a curing agent for SMUF resin, the temperature of curing declined to 104.1 °C, and the curing-induced heat released was as high as 495.7 mW/mg. This presents a strong proof that the SMUF resin could be completely cured at a relatively lower temperature in presence of CA-5. 

Integration of the pertaining curing peaks appearing in the DSC runs was undertaken and expressed as a function of time, so that the different curing systems could be easily differentiated (Figure 5). The times elapsed for completing of the curing reactions were, 10.6 min (CA-1), 11 min (CA-2), 10 min (CA-3), 10 min (CA-4), and 9.3 min (CA-5), respectively. Thus, the times needed to achieve a given degree of curing (50% is taken as an example) show the order CA-5 < CA-4 = CA-3 < CA-1 < CA-2. Interestingly, the curing rate of the CA-2-catalyzed SMUF resin is the lowest, which is not in line with its relatively low curing temperature and relatively large curing-induced heat release quantity assigned from DSC. However, this is thought to be related to the nature of the curing agent and its effect on the resin properties rather than to be attributed to the resin, which can be evidenced from the drop in viscosity, which accompanied the use of this curing system (Table 3).

### 3.4. Thermal Performance of the SMUF Resin Adhesive

Figure 6 and Table 4 show the TG curves and the corresponding degradation parameters of cured SMUF resin as a function of the employed curing system. It is obvious that all the degradation profiles are similar and followed the same trend. 

The DTG curves illustrate that the degradation traces can be divided into four main regions as follows: Region I: 50–220 °C, Region II: 220–270 °C, Region III: 270–400 °C and Region IV: 400–650 °C, which is clearly shown in Figure 6. The weight loss values are collected in Table 4.

The mass loss levels of CA-1 and CA-4 in region I are not considerably different while the mass loss in case of CA-5 was the lowest observed among this series. The loss in this temperature range is mainly attributed to steam and gas generation and volatilization of some micromolecular compounds that did not participate in the curing reaction. 

The loss in weight in region II for CA-1, CA-3, and CA-4 is not considerably different, whereas the loss level in case of CA-2 and CA-5 is much higher. Degradation of some polymeric chains as well as some unstable chemical bonds started to proceed in this phase. 

In region III, the corresponding weight loss for CA-1, CA-2, CA-3, CA-4, and CA-5 are the highest for a given sample: 38.79%, 32.34%, 35.44%, 29.59%, and 33.60%, respectively. The decomposition involved mainly severe rupture of the 3-dimensional network structure of the resin. 

There was very limited difference between the samples in the phase of region IV; the degradation encounters breakage of the principal peptide bonds of soy protein, and further decomposition of the network continues to generate intensive gases formation such as CO, CO_2_, NH_3_, and H_2_S. The existing of NH_3_ can affected the pH via produced various nitrogen byproducts.

The final residual weights in each case were 22.47%, 19.56%, 25.36%, 24.04%, and 28.34%, respectively. The lowest residual weight of was found for the sole use of (NH_4_)_2_HPO_4_ as a curing agent while it was the highest for the CA-5 curing system, which reflects a better thermal stability.

### 3.5. Interfacial Properties of SMUF Resin Adhesive under Different Curing Agents

Figure 7 shows the SEM images of fractured cross-sections of cured SMUF resin using the different curing systems. It reveals that the SMUF resin section could be mainly divided into three types: (1) Curing is incomplete or occurs so quickly that a loose, porous, and cracked resin is formed; (2) a dense section with a smooth surface is formed due to a full curing of the resin; and (3) curing is complete and accompanied by a section with crosslinking structures (seen from the red mark). 

The curing characteristics of (NH_4_)_2_SO_4_ are poor, and complete curing into a network structure is difficult to achieve. (NH_4_)_2_HPO_4_ is on the other hand a buffering-type acid. The section it cures from the resin is clean, and the cracking behavior is evidently improved. Thus, the (NH_4_)_2_HPO_4_-cured SMUF resin is closer to complete curing than the (NH_4_)_2_SO_4_-cured one. 

The loose and porous section of SMUF catalyzed by binary catalysts (CA-3 and CA-4) was improved to a considerable extent. When the ternary curing system was used (CA-5), the loose and porous structure of the resin section was replaced with a densely more compact crosslinking layer with considerably good hydrophobicity. This is clearly observed in case of CA-3, CA-4, and CA-5 but to a lesser extent in the case of CA-2. 

The crosslinking intertextures are almost evenly distributed in the case of CA-5. The traditional curing agents of amino resins mainly include acids (HCl and H_2_SO_4_), salts (NH_4_Cl and (NH_4_)_2_SO_4_), or latent curing agents. Curing is facilitated by direct or indirect H^+^ release, which promotes the polycondensation reaction required for curing. The crosslinking intertextures do not appear in the resins driven by these catalytic systems. However, the crosslinking intertextures were only realized for CA-2, CA-3, CA-4, and CA-5, where (NH_4_)_2_HPO_4_ was a common component of the employed curing systems. These results reveal that (NH_4_)_2_HPO_4_ plays somehow an important role in the crosslinking reaction of the resin, which may be demonstrated by Equations (7)–(11). When (NH_4_)_2_HPO_4_ was used alone as a curing agent, the curing temperature was low while the amount of liberated heat by the curing reaction was high. However, the thermal stability of the relevant system was very low on account of two additional processes promoted by the curing agent: H^+^ formation and acid generation (Equation (7)), which promote SMUF curing, and polyphosphoric acid formation, which results in the crosslinking effect in the SMUF system and esterification reactions with hydroxyl groups in the SMUF resin.
NH_4_HPO_4_ + HCHO → (CH_2_)_6_N_4_ + H_3_PO_4_ + H_2_O(7)
(8)$${\left({\mathrm{NH}}_{4}\right)}_{2}{\mathrm{HPO}}_{4}\overset{\Delta }{\to }2{\mathrm{NH}}_{3}+H_{3}{\mathrm{PO}}_{4}$$

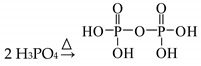
(9)

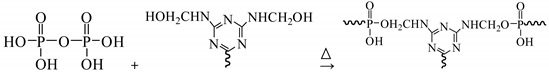
(10)

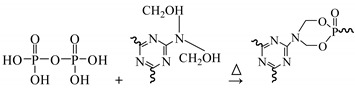
(11)

In case (NH_4_)_2_HPO_4_ was used as a principal curing agent, in combination with (NH_4_)_2_S_2_O_8_ and (NH_4_)_2_SO_4_, the two minor agents provide H^+^ ions needed for SMUF curing. The results of this study may serve as a reference for the curing of low-molar ratio amino resins applied to particleboard, plywood, medium-density fiberboard, and oriented strand boards.

## 4. Conclusions

The curing characteristics, thermal response, and interfacial adhesion of SMUF resin were found to be dependent on the type of employed curing agent. For instance, (NH_4_)_2_SO_4_ could not induce complete curing of SMUF resin, and the final shear strength of the resin was small while the water resistance was poor, and the interfacial adhesion section was loose and porous. With respect to (NH_4_)_2_HPO_4_, a high degree of crosslinking was reached in the resin system after hot press together with improved shear strength and water resistance, which can be attributed to a complicated decomposition behavior of the curing system. 

The use of a complex curing system incorporating (NH_4_)_2_HPO_4_ + (NH_4_)_2_SO_4_ + (NH_4_)_2_S_2_O_8_ involved significantly lowered curing temperature and maximized heat released by curing, and this contributed to a most favorable shear strength, resistance to hydrolytic degradation, and heat resistance of the resin.

## Figures and Tables

**Figure 1 polymers-11-01995-f001:**
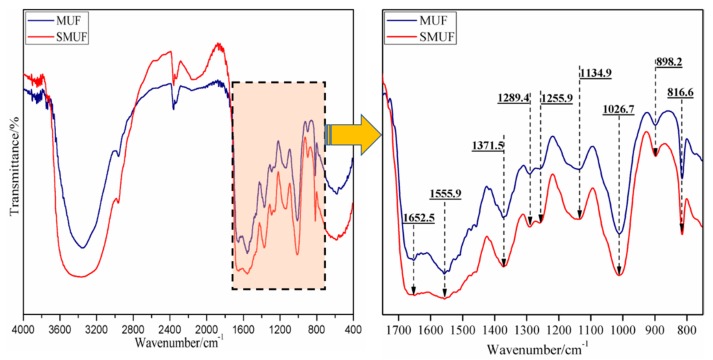
FT-IR spectra of melamine-urea-formaldehyde (MUF) and soybean protein-melamine-urea-formaldehyde (SMUF).

**Figure 2 polymers-11-01995-f002:**
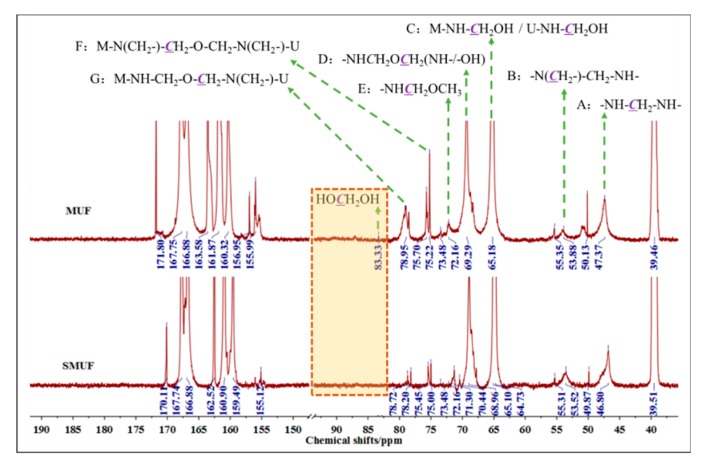
^13^C NMR spectra of MUF and SMUF.

**Figure 3 polymers-11-01995-f003:**
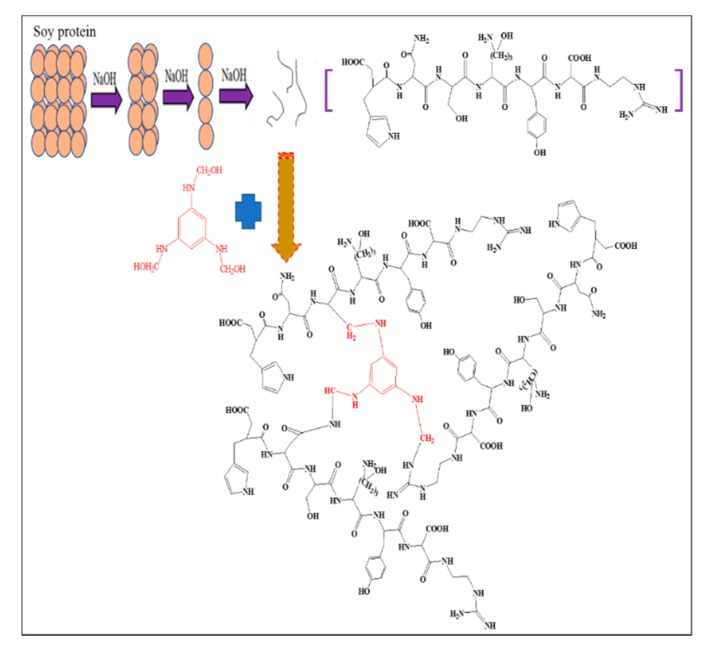
A suggestion for possible incorporation reaction of soy protein to MUF network.

**Figure 4 polymers-11-01995-f004:**
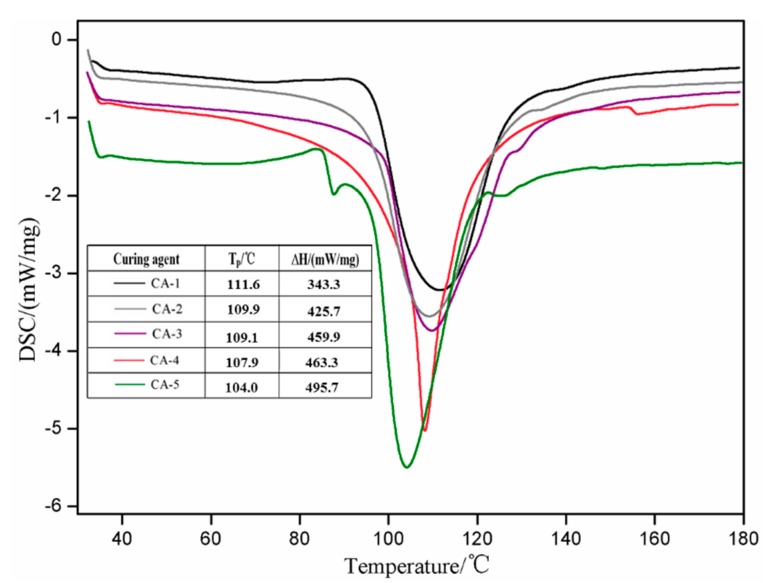
DSC thermograms of SMUF resins.

**Figure 5 polymers-11-01995-f005:**
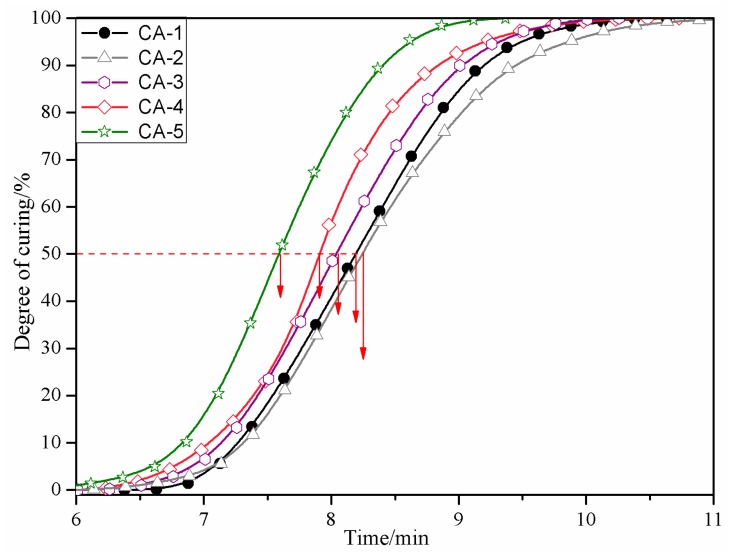
Developed degree of curing for SMUF using different curing agents as a function of time.

**Figure 6 polymers-11-01995-f006:**
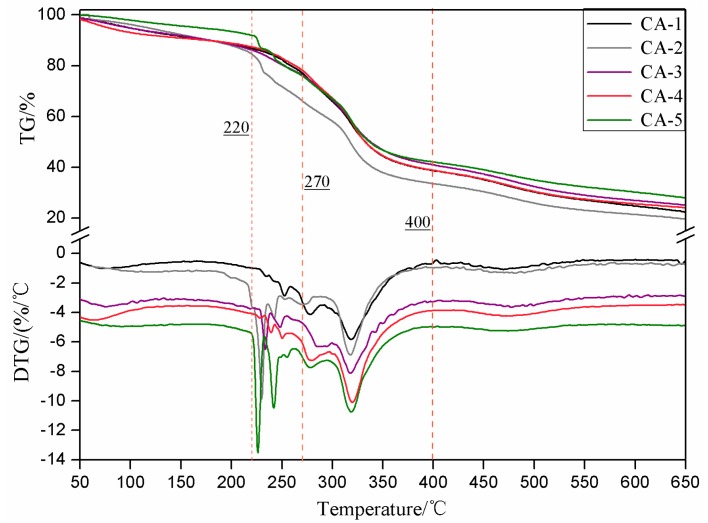
TG and DTG curves of SMUF resins cured with the different curing systems.

**Figure 7 polymers-11-01995-f007:**
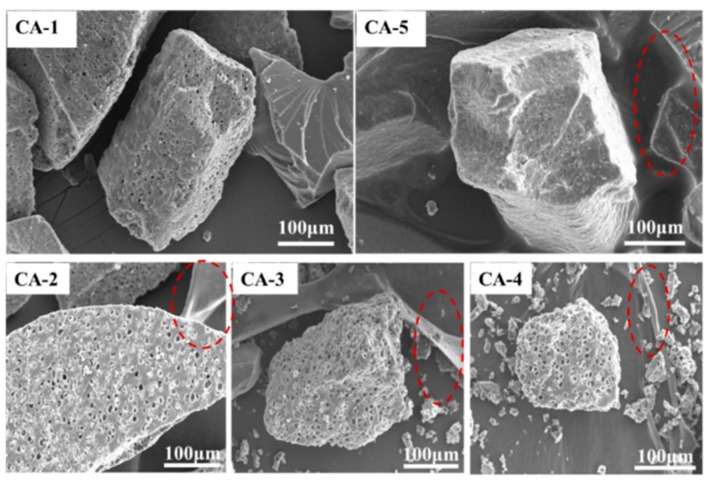
SEM images for cross-sections of fractured SMUF resins after curing with different curing agents.

**Table 1 polymers-11-01995-t001:** The composition of curing agent.

Curing Agent Abbreviation	Composition
CA-1	1.6% (NH_4_)_2_SO_4_
CA-2	1.6% (NH_4_)_2_HPO_4_
CA-3	1.2% (NH_4_)_2_HPO_4_ + 0.4% (NH_4_)_2_SO_4_
CA-4	1.2% (NH_4_)_2_HPO_4_ + 0.4% (NH_4_)_2_S_2_O_8_
CA-5	1.0 % (NH_4_)_2_HPO_4_ + 0.3% (NH_4_)_2_SO_4_ + 0.3% (NH_4_)_2_S_2_O_8_

**Table 2 polymers-11-01995-t002:** Properties of MUF and SMUF resins.

Resin Type	Viscosity/mPa·s	Solid Content/%	Appearance
MUF	590	66.7	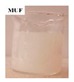
SMUF	725	67.2	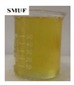

**Table 3 polymers-11-01995-t003:** Effect of the curing agents on the SMUF resin properties.

Curing Agents	Viscosity /mPa·s	Adhesive pH before Curing	Adhesive pH after Curing	Shear Strength in Boiling Water /MPa
Blank	1760	8.68	8.53	0.22 ± 0.04
CA-1	3400	7.17	6.43	1.71 ± 0.06
CA-2	1900	7.19	6.01	2.03 ± 0.09
CA-3	2350	7.01	6.13	2.16 ± 0.11
CA-4	2050	6.32	5.61	2.31 ± 0.08
CA-5	3150	7.03	6.18	2.58 ± 0.09

**Table 4 polymers-11-01995-t004:** TG parameters of the SMUF resin cured with different curing systems.

Curing Agents	Temperature Interval/°C	Weight Loss/%	Char Yield/%
CA-1	50–220	13.18	22.47
	220–270	9.30	
	270–400	38.79	
	400–650	16.26	
CA-2	50–220	15.57	19.56
	220–270	18.52	
	270–400	32.34	
	400–650	14.01	
CA-3	50–220	13.81	25.36
	220–270	9.86	
	270–400	35.44	
	400–650	15.53	
CA-4	50–220	12.63	24.04
	220–270	9.06	
	270–400	29.59	
	400–650	14.68	
CA-5	50–220	8.00	28.34
	220–270	16.19	
	270–400	33.60	
	400–650	13.87	

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
