# Peer review of "Influence of Single/Collective Use of Curing Agents on the Curing Behavior and Bond Strength of Soy Protein-Melamine-Urea-Formaldehyde (SMUF) Resin for Plywood Assembly"

_polymers, 2019, doi:10.3390/polym11121995_

Round 1
Reviewer 1 Report
The readability of the graphs should be solved. And there is a lack of debate based on the results of the experiment. 1. For soybean protein hydrolysates, it is appropriate to indicate basic information, even if there is a reference. 2. Tree species must be specified accurately. 3. Directly specifying the F / (M + U) ratio is not appropriate. 4. Show the mixing conditions in a table 5. You can use Chinese National Standard under test conditions, but I think it is appropriate to specify international standards such as ISO. 6. What exactly is the structure of soybean protein hydrolysate? I know that SPH can form various structures in some cases. 7. Is the structure of Soybean protein hydrolysate defined in Figure 3? 8. I would like to reorganize the possible reactive structures in Figure 3. It is inconvenient to understand. 9. Why is there no basic adhesive strength in the characteristic analysis by curing conditions? 10. Add the sample condition of DSC to the experiment part. 11. It is difficult to see the exact graph of DSC. Redraw it as a Y axis-offset. 12. Why did you decide on a Degree of curing of 50%? 13. In conclusion, can't we predict that the difference in strength is caused by the difference in degree of hardening? In other words, can you control the degree of hardening to achieve similar results? 14. In Figure 6, the graph is not distinct. 15. Since ammonium phosphate is used, ammonia is generated and according to this phenomenon, have you not considered secondary reaction induction and pore generation?Author Response
Response to Reviewer 1 Comments:
The readability of the graphs should be solved. And there is a lack of debate based on the results of the experiment.
Response: The readability has been improved, and provided deeply discussion.
For soybean protein hydrolysates, it is appropriate to indicate basic information, even if there is a reference.Response 1: The viscosity and solid content of the hydrolyzed soybean protein are added and the used method has been cited [reference 16].
Tree species must be specified accurately.Response 2: Thank you for your suggestion, the tree species (poplar wood with latin-name) was provided in section 2.1. in order to specify the wood species (please check the red text in this section).
Directly specifying the F / (M + U) ratio is not appropriate.Response 3: Thank you for your suggestion, this sentence was changed to “SMUF resin adhesive was prepared as follows:……” (please check the first sentence in section 2.2.).
Show the mixing conditions in a tableResponse 4: Thank you for your suggestion, we preferred to provide all the mixing conditions in the text of parts 2.2 and 2.3.
You can use Chinese National Standard under test conditions, but I think it is appropriate to specify international standards such as ISO.Response 5: Thank you for your advice, but please note that all the measurements of this work have been performed according to Chinese National Standards, however, we will consider this in our future work.
What exactly is the structure of soybean protein hydrolysate? I know that SPH can form various structures in some cases.Response 6: It is hard to say the exactly structure of soybean protein hydrolysate because it is a complicated structure with -COOH and -NH2 terminal groups. Further, our interest in this material is based on its chemical functionality and the hydrolysis conditions. In all cases, reference [24] was added in the text.
Is the structure of Soybean protein hydrolysate defined in Figure 3?Response 7: No, Fig. 3 just provides a general postulation of possible incorporation reaction of soy protein to MUF network
I would like to reorganize the possible reactive structures in Figure 3. It is inconvenient to understand.Response 8: Sorry for this, we did reorganize the possible reactive structure in Figure 3 to make it clearer.
Why there is no basic adhesive strength in the characteristic analysis by curing conditions?Response 9: In this work, our main target is that using the single/collective curing agents to investigate the effect of curing agent on the properties of wood adhesives. Thus, the basic information on adhesive strength was neglected in the current manuscript, please refer to our previous report (reference 16).
Add the sample condition of DSC to the experiment part.Response 9: We added the missed sample conditions in section 2.7 (marked in red).
It is difficult to see the exact graph of DSC. Redraw it as a Y axis-offset.Response 11: Actually, the main peaks can be easily distinguished in Fig. 4 while the other peaks are trivial and can be ignored.
Why did you decide on a Degree of curing of 50%?Response 12: I am very sorry for making you confused. Actually, 50% is just taken as an example, other people also can choose different degrees of curing.
In conclusion, can't we predict that the difference in strength is caused by the difference in degree of hardening? In other words, can you control the degree of hardening to achieve similar results?Response 13: Theoretically, very close results can be achieved using the same processing, considering the same degree of curing was obtained.
In Figure 6, the graph is not distinct.Response 14: You are right, it is not distinct, we have interpreted in the text of part 3.4.
Since ammonium phosphate is used, ammonia is generated and according to this phenomenon, have you not considered secondary reaction induction and pore generation?Response 15: we kept this consideration in mind as you said, please take a look on Figure 7 and the related explanation.
Reviewer 2 Report
please see the attached file

Author Response
Response to Reviewer 2 Comments:
you probably mean curing !!!Response 1: Yes, this is a spelling mistake, It is now corrected to “curing”.
The terms resin and adhesives are too much. please omit the adhesivesResponse 2: Thank you for your suggestion, the word “adhesives” is removed.
This is a very poor introduction, with generalisations. I would like to see references to be directly related to the topic and especially to the resins applied nowadays in the plywood industry. There are many !!!Response 3: Thank you for your suggestion, we dealt with this problem. This part has been reorganized and improved now, please check in the manuscript.
Inappropriate style of references. This comment applies to the whole paper.Response 4: We changed the references style through the whole manuscript according to the author guidelines.
reference for this important observation?Response 5: Related reference for this important observation [ref. 20] was cited and added, please check in the manuscript.
number of replicates? please specifyResponse 6: the number of replicates have been specified in appropriate part (section 2.4, marked in red), please check "...... Triplicate of three-layer plywood......".
The session 3 is just a presentation of the results obtained in this study, which indeed are useful. However, there is a huge luck of scientific discussion in this session. The authors did not mention any similar studies conducted in the past and just focused themselves on their results. To me this paper, is more like a technical report prepared for an industry rather than a scientific paper.Response 7: Thank you for your suggestion, this part has been improved and some reports were included to compare our results with previous studies, please check section 3.4 after re-organization, along with the added relevant references.
Round 2
Reviewer 1 Report
I think it has been organized in many parts. But there are still some things to fix. Readability 1. The five sampling conditions of Line 93 ~ 95 are not clear. Make it a separate table. 2. The DSC graph in Figure 4 is not clearly distinguishable. The same is true for Figure 6. Change the line color and line shape. Research contents 1. Does the condition of 130 degrees 5 minutes form the same curing rate for each curing agent? If not, can it be interpreted that there is a difference in strength by degree of curing? If it is correct, it can be interpreted that different hardening structure is formed for each curing agent or second effect is formed. 2. The second reaction of NH3 means that it can produce various nitrogen by-products and this can affect pH, etc. 3. The basic adhesive strength I mentioned is before water boiling. This is because I want to see how much the adhesive strength decreases through the WB test.Author Response
I think it has been organized in many parts. But there are still some things to fix.
Readability
1. The five sampling conditions of Line 93 ~ 95 are not clear. Make it a separate table.
Response 1: Thank you for your suggestion, we have now added a new table as “Table 1” instead of those texts, please check the blue text, meanwhile, the other table was updated.
2. The DSC graph in Figure 4 is not clearly distinguishable. The same is true for Figure 6. Change the line color and line shape.
Response 2: Figure 4 and Figure 6 have both made improved now, and the readability was increased via Y axis-offset for DSC in Figure 4 and DTG in Figure 6.
Research contents
1. Does the condition of 130 degrees 5 minutes form the same curing rate for each curing agent? If not, can it be interpreted that there is a difference in strength by degree of curing? If it is correct, it can be interpreted that different hardening structure is formed for each curing agent or second effect is formed.
Response 2: Actually, 130 ℃ and 5 min is hot pressed temperature and time, respectively, for all plywood manufacturing. This leads to the different curing and thermal behaviors as well as the interfacial properties, the detailed interpretation was shown in section of 3.2-3.5.
2. The second reaction of NH3 means that it can produce various nitrogen by-products and this can affect pH, etc.
Response 2: Exactly, the text have been improved marked in blue text (section 3.4).
3. The basic adhesive strength I mentioned is before water boiling. This is because I want to see how much the adhesive strength decreases through the WB test.
Response 3: Thank you for your question, inversely, the water boiling properties of wood adhesives was increased compared to basic adhesive, author supplemented the experiment and added the value in Table 3, please check it in section of 3.2.
Reviewer 2 Report
Dear authors
The paper is greatly improved, the authors took into their consideration the comments and therefore I am happy to suggest publication in its present form
Author Response
1. The paper is greatly improved, the authors took into their consideration the comments and therefore I am happy to suggest publication in its present form
Response 1: Thank you for your comment and approval.
Round 3
Reviewer 1 Report
Thank you for your kind reply. The main parts seem to have been modified very well. We look forward to further research on the efficiency of each additive under various environmental parameters. Expect good research results.